# Contrasting impacts of two types of El Niño events on winter haze days in China's Jing-Jin-Ji region

Xiaochao Yu[1,2], Zhili Wang[1*], Hua Zhang[1], Jianjun He[1], Ying Li[3]

[1]State Key Laboratory of Severe Weather and Key Laboratory of Atmospheric Chemistry of CMA, Chinese Academy of Meteorological Sciences, Beijing, 100081, China

[2]Department of Atmospheric and Oceanic Sciences, Fudan University, Shanghai, 200438, China.

[3]National Climate Center, China Meteorological Administration, Beijing, 100081, China

*Correspondence to*: Zhili Wang (wangzl@cma.gov.cn)

**Abstract.** El Niño is complicated due to its diverse distribution features and intensities. The regional climate anomalies caused by different types of El Niño event likely lead to various impacts on winter haze pollution in China. Based on long-term site observations of haze days in China from 1961 to 2013, this study explores the effects of Eastern Pacific (EP) and Central Pacific (CP) types of El Niño events on the number of winter haze days (WHD) in China's Jing-Jin-Ji (JJJ) region and the physical mechanisms underlying WHD changes. The results show statistically significant positive and negative correlations, respectively, between WHD in the JJJ region and EP and CP El Niño events. At most sites in the JJJ region, the average WHD increased in all EP El Niño years, with the maximum change exceeding 2.0 days. Meanwhile the average WHD decreased at almost all stations over this region in all CP El Niño years, with the largest change being more than -2.0 days. The changes in large-scale circulations indicate obviously positive surface air temperature (SAT) anomalies and negative sea level pressure (SLP) anomalies over North China, and southerly wind anomalies at the mid-low troposphere over eastern China in the winters of EP El Niño years. These anomalies are conducive to increases in WHD in the JJJ region. However, there are significant northerly and northwesterly wind anomalies at the mid-low troposphere over eastern China, and stronger and wider precipitation anomalies in the winters of CP El Niño years, which contribute to decreased WHD over the JJJ region. Changes in local synoptic conditions indicate negative SLP anomalies, positive SAT anomalies, and weakened northerly winds over the JJJ region in the winters of EP El Niño years. The total occurrence frequency of circulation types conducive to the accumulation (diffusion) of aerosol pollutants is increased (decreased) by 0.4% (0.2%) in those winters. However, the corresponding frequency is decreased (increased) by 0.5% (0.6%) in the winters of CP El Niño years. Our study highlights the importance of distinguishing the impacts of these two types of El Niño events on winter haze pollution in China's JJJ region.

## 1 Introduction

North China, with the Jing-Jin-Ji (JJJ) region at the core, has encountered continuous severe haze pollution in recent winters. These atmospheric calamities have seriously harmed traffic, economic development, and resident health in this region (Gao et al., 2017; Liu et al., 2017; Zhang et al., 2019a). Increased anthropogenic emissions are considered the predominant reason for the increased frequency and intensity of haze pollution. However, many studies have verified the effects of worsening local weather conditions caused by large-scale climatic anomalies on severe haze events (Li et al., 2016; Cai et al., 2017; Li et al., 2018; Yin and Wang, 2018). Anomalous meteorological conditions have significant influences on the development and maintenance of haze events; in particular, the explosive increase in local air pollutants is always accompanied by anomalous atmospheric circulation conditions (He et al., 2018; Zhang et al., 2018; Zhong et al., 2018). Hence, identifying the mechanism underlying the response of haze events to worsening weather conditions caused by interannual climate changes has implications for effectively controlling haze pollution and improving air quality.

As a strongest signal of interannual climate variation (Wyrtki, 1975; Cane, 2005), El Niño has an important influence on the maintenance and diffusion of air pollutants via affecting large-scale atmospheric circulation and precipitation (Feng et al., 2016a, 2016b; Zhao et al., 2018), and consequently modulates the interannual variation of winter haze days (WHD) in China (Gao and Li, 2015; Sun et al., 2018; He et al., 2019). Several studies have reported that an anomalous anticyclone develops over the Northwest Pacific during the maturation of El Niño, resulting in increased precipitation and decreased WHD in southern China (Li et al., 2017; Zhao et al., 2018; He et al., 2019). Moreover, atmospheric circulation anomalies caused by El Niño can exacerbate the northward transport of aerosols in South and Southeast Asia, thereby increasing winter mean aerosol concentrations (Feng et al., 2016a) and intraseasonal severe haze days in eastern China (Zhao et al., 2018; Yu et al., 2019). Recent studies have indicated that there is a significant negative correlation between El Niño and WHD in southern China (Li et al., 2017; Zhao et al., 2018; He et al., 2019). However, the impacts of El Niño on WHD in northern China remain controversial. For example, Sun et al. (2018) showed that El Niño led to increased WHD in North China by suppressing the activity of the East Asian winter monsoon (EAWM). However, based on statistical analyses of long-term site observations of WHD in China, several studies have found no statistically significant correlation between El Niño indices and WHD in North China (Li et al., 2017; Zhao et al., 2018; He et al., 2019).

The above studies mostly focused on analyzing the comprehensive impacts of all El Niño events on WHD in China. Their results indicated that the effect of El Niño events on air pollutants in northern China was much weaker than that in southern China (Li et al., 2017; Zhao et al., 2018; He et al., 2019). However, the El Niño-Southern Oscillation (ENSO) is a complex system with two dominant modes of quasi-quadrennial and quasi-biennial oscillations coexisting in the tropical Pacific (Bejarano et al., 2008; Wang and Ren, 2017). Its warm conditions (El Niño) can be classified into the Eastern Pacific (EP) and Central Pacific (CP) El Niño according to the anomalous sea surface temperature (SST) patterns contributed by the interplay of these independent modes (Ashok et al., 2007; Levine et al., 2010; Roberts et al., 2016; Timmermann et al., 2018). Because of the significantly distinct SST anomaly patterns in the equatorial Pacific, the two types of El Niño events have different influences on the Walker circulation, which further stimulates global circulation wave trains and results in contrasting temperature and precipitation anomalies in East Asia (Larkin et al., 2005; Yuan et al., 2012; Cai et al., 2018). The anomalies in regional climate caused by the two types of El Niño events may have different influences on winter atmospheric pollutants in China. For example, using the tropospheric chemical model GEOS-Chem, Feng et al. (2016a) showed that CP El Niño played an important role in redistributing seasonal mean $PM_{2.5}$ (particulate matter with a diameter $\leq$ 2.5 μm) concentrations in China. Recently, Yu et al. (2019) also found significant opposite changes in winter mean aerosol concentrations and severe haze days in North China in the responses to different types of El Niño events by using a global aerosol-climate model. Nevertheless, the observation-based studies on the effects of the two types of El Niño events on haze pollutants in China are still insufficient. The JJJ region is one of the most densely populated areas in China and a typical region of severe air pollution (Cai et al., 2017; Miao et al., 2017; Zhong et al., 2018). Therefore, it is important to understand the different responses of WHD in this region to the two types of El Niño events in greater depth.

This study first classifies different types of El Niño events according to the latest national standard of the People's Republic of China (PRC) "Identification method for El Niño/La Niña events" issued by the China Meteorological Administration (CMA) (Ren et al., 2017). Then, we explore the impacts of the two types of El Niño events on WHD in China's JJJ region (37–42°N, 113–120°E) from the perspectives of large-scale circulation and local synoptic condition anomalies using long-term site observations and reanalysis datasets, combined with commonly used circulation type classification methods. The datasets and methods used in this study are presented in Section 2. The impacts of the two types of El Niño events on WHD in China's JJJ region and the potential physical mechanisms are analyzed in Section 3. The discussion and conclusions are presented in Section 4.

## 2 Methodology

### 2.1 Data

The datasets used in this study were as follows. (1) The monthly haze days dataset from the National Meteorological Information Center of the CMA. The time span of the dataset is from March 1961 to February 2013. According to a comprehensive judgment method widely used in previous studies, a haze day is identified when the daily mean visibility is less than 10 km and the daily mean relative humidity is less than 90% (Schichtel et al., 2001; Doyle et al., 2002; Wu et al., 2010). (2) The monthly Niño3 index (SST anomaly averaged over the Niño3 domain [150°W–90°W, 5°S–5°N]; $I_{Niño3}$), Niño4 index (same as the Niño3 index, but over the Niño4 domain [160°E–150°W, 5°S–5°N]; $I_{Niño4}$), and Niño3.4 index (same as the Niño3 index, but over the Niño3.4 domain [170°W–120°W, 5°S–5°N]; $I_{Niño3.4}$) from March 1961 to February 2013, provided by the National Climate Center of the CMA. All Niño indices are calculated using the Hadley Centre Sea Ice and Sea Surface Temperature Data (HadISST) from March 1961 to December 1981 and the National Oceanic and Atmospheric Administration (NOAA) daily optimum interpolation (OI.v2) SST dataset from January 1982 to February 2013 (Ren et al., 2017). (3) Daily and monthly ERA-40 and ERA-Interim reanalysis data from the European Centre for Medium-range Weather Forecasts (ECMWF), including sea level pressure (SLP), air temperature at 2 m, wind at 10 m, geopotential height at 500 hPa, and wind from 1000 to 850 hPa (composed of seven pressure levels at 850, 875, 900, 925, 950, 975, and 1000 hPa). The horizontal resolution is 0.25°×0.25°, and the time span is from March 1961 to February 2013 for both daily and monthly reanalysis data. The data from March 1961 to December 1978 are derived from the ERA-40 reanalysis data, and the data from January 1979 to February 2013 are derived from the ERA-Interim reanalysis data. (4) The global land surface precipitation data were provided by the Global Precipitation Climatology Centre (GPCC), with a horizontal resolution of 0.5°×0.5°, covering March 1961 to February 2013 (Schneider et al., 2014).

### 2.2 Identification of two types of El Niño events and calculation of their indices

Similar to Yu et al. (2019), we classified different types of El Niño events referring to the national standard of the PRC mentioned in Section 1. This method identifies El Niño events based on the widely used $I_{Niño3.4}$ and employs $I_{Niño3}$ and $I_{Niño4}$ to distinguish the different characteristics of the two types of El Niño events. $I_{Niño3}$ and $I_{Niño4}$ are highly sensitive to EP and CP El Niño events, respectively. This identification method has been applied to the climate operations of the CMA and has been widely used in research on the effects of El Niño events (e.g., Mu et al., 2017; Yu et al., 2019). We first selected all El Niño events from 1961 to 2013. An El Niño event is identified when the absolute value of the 3-month smoothing average of $I_{Niño3.4}$ reaches or exceeds 0.5°C for at least 5 months. All El Niño events were classified referring to the EP El Niño index (Iep) and the CP El Niño index (Icp). Iep and Icp were calculated as follows:

$$Iep = I_{Niño3} - (\alpha \times I_{Niño4}), \tag{1}$$

$$Icp = I_{Niño4} - (\alpha \times I_{Niño3}). \tag{2}$$

According to an empirical formula, the constant $\alpha$ is 0.4 if $I_{Niño3} \times I_{Niño4} > 0$, but 0 if $I_{Niño3} \times I_{Niño4} \leq 0$. An event is defined as an EP (CP) El Niño event if the absolute value of Iep (Icp) reaches or exceeds 0.5°C for at least 3 months. Table 1 shows the specific classifications of the two types of El Niño events obtained by the above method.

### 2.3 Circulation type classification methods

An aim of using circulation type classification is to identify the most frequently occurring subset of the meteorological data, thereby considering the numerous interrelated meteorological variables within an integrated framework and exploring the physical mechanisms underlying aerosol pollution in the JJJ region in the classification process (Richman et al., 1981; Miao et al., 2017). Among the multitudinous circulation classification techniques, the T-mode principal component analysis (PCA) combined with the K-mean cluster used in this study is the most effective identification approach because of its reproduction

of predefined types, temporal and spatial stability, and low dependence on preset parameters (Huth et al., 1996; Zhang et al., 2012). This method has been widely used to identify the circulation types associated with air pollution (He et al., 2017a, 2017b, 2018). Similar to He et al. (2018), daily SLP data from March 1961 to February 2013 in the JJJ region were taken as the samples for circulation type classification. First, we reshaped three-dimensional daily SLP data, including time, latitude, and longitude, into two-dimensional data (time × grid) and normalized the two-dimensional data for time series. Second, the normalized SLP data performed the T-mode PCA and its main components were obtained according to the cumulative variance contributions, up to a total of 95%. Third, we clustered the main components using the K-means cluster and identified the optimal number of clusters referring to the criterion function (Liu and Gao, 2011). In this study, the inflection point of the criterion function, which represents the optimal number of clusters, was eight. The daily SLP data were assigned to eight synoptic-scale circulation types based on the clustering result. The other variables (e.g., temperature at 2 m and wind at 10 m) were classified in the same way. Finally, each pattern of synoptic-scale circulation was determined.

## 2.4 Correlation analysis

The correlation coefficients of site-observed WHD in eastern China (east of 110°E) with the different types of El Niño indices (i.e., $I_{Nino3.4}$, Iep, and Icp) were calculated in this study. The sites without WHD for at least 25 consecutive years were eliminated before the correlation analysis, as the time series of WHD at these sites lack interannual and interdecadal fluctuations, and their responses to anomalous synoptic conditions caused by climate change are weak. In addition, a band-pass filtering of 2–10 years was performed for the WHD data to remove signal interference from changes in local aerosol emissions and interdecadal climate variability following Zhao et al. (2018) and He et al. (2019). The final results more intuitively reflect the correlation between El Niño events and WHD.

## 3 Results

### 3.1 Impacts of the two types of El Niño events on WHD in China's JJJ region

Figure S1 shows the correlation coefficients for the time series of site-observed WHD in eastern China and the $I_{Nino3.4}$, Iep, and Icp indices. Whether for EP or CP El Niño events, the indices feature a uniformly negative correlation with WHD at most of the stations in southern China. This result is in agreement with previous studies (e.g., Li et al., 2017; Zhao et al., 2018; He et al., 2019), which reported that the increase in precipitation over southern China due to the anomalous anticyclone over the West Pacific during the mature phase of El Niño events significantly reduced WHD in this region. However, the sign of the correlation coefficient between EP El Niño and WHD is completely opposite that between CP El Niño and WHD for the majority of sites in the JJJ region. For most sites, WHD is positively correlated with the Iep index (121 sites, accounting for 62.1% of all sites) but negatively correlated with the Icp index (126 sites, 64.6%). Considering the various inducements of haze pollution, such as local emissions, weather conditions, and topography, there may be some biases in correlation coefficients among different sites when examining the correlations between the single-site WHD and El Niño indices. However, the consistent positive and negative correlations at more than 60% of stations in response to EP and CP El Niño events support the opposite impacts of the two types of El Niño events on WHD in the JJJ region. The corresponding proportions increase to 70.5% and 86.2%, respectively, if we only count the stations where the correlations pass a significance level of 90%. As seen in Figure 1, the absolute values of the correlation coefficients at some stations exceed 0.4. There are statistically significant correlations between the site-averaged WHD in the JJJ region and the Iep and Icp indices ($p \leq 0.05$), with correlation coefficients of 0.16 and –0.2, respectively (Table 2). These low correlation values likely imply a mild impact of ENSO on WHD. However, the corresponding correlation coefficients reach 0.31 and –0.43, respectively, with a confidence level of 99%, when only considering the stations at which the correlations pass a significance

level of 90% (Fig. 1).

Figure 2 shows the composite anomalies of WHD at all sites over the JJJ region in different types of El Niño years relative to the 1961–2013 mean WHD. For the majority of stations in the JJJ region, WHD is increased in EP El Niño years (149 stations, 76.4% of all stations), with the maximum change exceeding 2.0 days (accounting for 17–79% of the average WHD; Fig. S2b). However, WHD is reduced at almost all stations over this region in CP El Niño years (172 stations, 91.8% of all stations), with the maximum change exceeding –2.0 days (accounting for –13% to –70% of the average WHD; Fig. S2c). For instance, in EP El Niño years, there are significant increases in WHD surrounding Beijing and Tianjin, in which the positive anomalies generally exceed 1.2 days. In CP El Niño years, the comparable negative WHD anomalies can be seen in the same region. The opposite differences in WHD corresponding to the two types of El Niño events are also apparent in the northwestern and northeastern parts of the JJJ region. The spatial correlation coefficient between the anomalous WHD in the JJJ region in both types of El Niño years reaches –0.71, which is significant at the 99% level.

The detailed statistics of WHD anomalies at all sites over the JJJ region in each El Niño year are shown by the box-and-whisker plots in Figure 3. As mentioned above, the WHD variations in the JJJ region are disturbed by local emissions, weather conditions, and topography. These result in a spread of distributions of WHD anomalies in response to individual EP or CP El Niño years. As seen in Figure 3a, the medians of WHD anomalies for all sites are below the zero line in all CP El Niño years, indicating a negative WHD anomaly for more than half of the sites. Although the medians of WHD anomalies fluctuate above and below the zero line in different EP El Niño years, the anomalies of WHD show obviously wider distributions in the positive range for all sites in each year, with the positive extremum exceeding 10 days in most EP El Niño years (data not shown in Fig. 3a). In addition, the distributions of WHD anomalies in different types of El Niño years also display interdecadal variations. The quasi-quadrennial mode was significantly strong, and EP events occurred more frequently during 1980–1999, corresponding to a larger proportion and higher extremum of positive WHD anomalies for all sites in the JJJ region. After 2000, the frequency of CP El Niño events was increased corresponding to the dominant quasi-biennial mode in the tropical Pacific, which led to a larger proportion and higher extremum of negative WHD anomalies in the JJJ region. This phenomenon may be attributable to the interdecadal transformation of the relative activity or stability between the two types of El Niño modes (Wang and Ren, 2017). Figure 3b also shows that the WHD anomalies are mainly located in the positive range in the EP El Niño years, but are obviously located in the negative range in the CP El Niño years.

In summary, the impacts of the two types of El Niño events on WHD are clearly opposite over the JJJ region. The EP El Niño events lead to increases in WHD in the JJJ region, whereas the CP El Niño events decrease WHD in this region. This is the reason why the correlation between the time series of WHD over North China and the El Niño indices was found to be statistically insignificant when considering the El Niño events as a whole in previous studies (e.g., Li et al., 2017; Zhao et al., 2018; He et al., 2019).

## 3.2 Anomalies of winter mean large-scale circulations for two types of El Niño

Next, we explore the physical mechanisms underlying the WHD changes in the JJJ region in response to EP and CP El Niño events from the perspective of large-scale circulation anomalies (Fig. 4). Previous studies have found that the severe haze events over North China in boreal winter were always accompanied by a decrease in northerly wind speed in the lower troposphere and weakening of the East Asian trough in the middle troposphere (Chen and Wang, 2015). The formation of heavy haze pollution over Beijing and its surroundings is significantly facilitated by the weakened EAWM, high-pressure anomalies at 500 hPa, and enhanced atmospheric stability (Zhang et al., 2014; Zhong et al., 2018).

The surface air temperature (SAT) generally increases over East Asia in the winters of EP El Niño years, especially in northern China, northeastern China, and eastern Siberia, with the maximum increase reaching 2 K (Fig. 4a). The SLP generally drops over East Asia. In particular, the SLP is decreased more significantly north of 30°N, with the maximum reaching –4 hPa in eastern Siberia (Fig. 4b). On the one hand, the worsening meteorological conditions, including near-

surface warming and low pressure, are not conducive to the southward movement of the Siberian high pressure system, thereby weakening the transport of the EAWM on aerosol pollutants over northern China. On the other hand, such conditions promote relatively stable circulation, which is conducive to the accumulation of aerosol pollutants. In addition, there is a significant positive anomaly of geopotential height at 500 hPa over the northwestern Pacific in the winters of EP El Niño years, with the maximum anomalies exceeding 20 gpm over southern Japan and the northwestern Pacific. These positive geopotential height anomalies also extend westward over northeastern and eastern China (Fig. 4c). At the same time, there is a negative geopotential height anomaly at 500 hPa over southwestern China. Consequently, such distribution of geopotential height anomalies results in an anomalous southerly wind in the middle and lower troposphere over northeastern and eastern China (Fig. 4d). The anomalous southerly wind weakens the seasonal prevailing northwesterly wind in the JJJ region, with the maximum decrease exceeding 0.5 m s$^{-1}$. This type of large-scale circulation anomaly suppresses the outward transport of aerosol pollutants in this region. Similar circulation anomalies were also found during the 2015/2016 super-strong EP El Niño event in an earlier study (Chang et al., 2016).

Compared to the EP El Niño years, there are larger increases in SAT and decreases in SLP over southern China in the winters of CP El Niño years, with the maximum changes reaching 0.8 K and –3 hPa, respectively, over the south of the Yangtze River (Fig. 4f and g). However, the positive SAT anomalies and negative SLP anomalies over northern China in the winters of CP El Niño years are apparently weaker than the corresponding changes in the winters of EP El Niño years (Fig. 4f and g). The SAT is significantly decreased in northeastern China and Siberia, with the largest negative anomalies reaching –2 K. Additionally, there is an anomalous negative geopotential height at 500 hPa over west of Lake Baikal and the Aleutian region, but a positive geopotential height at 500 hPa over southern Japan and the Korean peninsula in the winters of CP El Niño years (Fig. 4h). This leads to the westward shift of the East Asian trough (Jiang et al., 2017). As a result, there are northerly and northwesterly wind anomalies in the middle and lower troposphere north of 30°N in China, which significantly enhances the seasonal prevailing northerly wind (Fig. 4i). Such anomalous circulations are conducive to the outward transport of aerosol pollutants in the JJJ region. The monthly mean precipitation is significantly increased over eastern China in the winters of CP El Niño years, especially in the coastal regions of southeastern China, with the maximum changes exceeding 20 mm. Precipitation is generally increased over southern China, with the maximum changes exceeding 10 mm, but decreased slightly over central and northeastern China in the winters of EP El Niño years. The range of anomalous positive precipitation over the JJJ region is wider in CP El Niño years compared to that in EP El Niño years, although a comparable increase in precipitation over this region occurs with both types of El Niño years (Fig. 4e and j). Thus, the former is more conducive to enhancing the wet deposition of particulate matter.

Previous studies have emphasized the significant contributions to haze pollution of the formation of secondary inorganic and organic aerosols (Huang et al., 2014; Cheng et al., 2016; Wang et al., 2016a). Ma et al. (2017) attributed the elevation of PM$_{2.5}$ from heavy (150–250 µg m$^{-3}$) to severe (>250 µg m$^{-3}$) pollution to aerosol chemical conversion processes, which dominate the later stages of severe haze pollution. According to chamber studies and ambient measurements, the formation of secondary aerosols and their physical and chemical characterizations are markedly dependent on both temperature (Warren et al., 2009; Ding et al., 2011; Clark et al., 2016) and relative humidity (RH; Liu et al., 2011; Nguyen et al., 2011; Sun et al., 2013; Li et al., 2018). Given the lower temperature and ozone concentrations and higher coal consumption in winter in northern China (Chen et al., 2015), heterogeneous reactions related to sulfate and nitrate, rather than photochemical reactions, are considered mostly responsible for the increased PM$_{2.5}$ concentrations (Ma et al., 2017). This links the higher PM$_{2.5}$ concentrations with higher RH, as that factor contributes to these heterogeneous reactions (Cheng et al., 2016; Wang et al., 2016a). However, some studies have found nonsignificant or negative correlations between RH and WHD over northern China (Chen and Wang, 2015; Wu et al., 2016; He et al., 2019). Our results show apparently positive RH anomalies over eastern China in the winters of CP El Niño years. By contrast, increases in RH mainly occur over southern China, and RH is slightly increased or decreased over the JJJ region in the winters of EP El Niño years (Fig. S3). The changes in RH in the

mid-low troposphere over northern China in response to EP and CP El Niño years are not consistent with the corresponding variations in WHD in the JJJ region. This indicates that the regional transport of aerosol pollutants dominates the variations of WHD in the JJJ region in response to the two types of El Niño events, which supports the situation at the initial stage of haze occurrence as reported in Ma et al. (2017).

### 3.3 Anomalies in intraseasonal local synoptic conditions in the winters of different types of El Niño years

In this section, we further explore the different effects of the two types of El Niño events on WHD in the JJJ region from the perspective of changes in intraseasonal local synoptic conditions. Using the T-mode PCA and K-means cluster analysis methods, eight circulation types were identified over the JJJ region in winter. The effects of the two types of El Niño events on these circulation types were then compared. The changes in local synoptic conditions are defined as the differences between the results averaged in 10 EP (6 CP) El Niño years and the climatology.

Figures S4 and S5 show the climatological distributions of SLP, air temperature at 2 m, and wind at 10 m, respectively, over the JJJ region in winter for the eight circulation types. A larger northwest-southeast SLP gradient (Fig. S4a, b, c, and d) and a stronger northerly wind (Fig. S5a, b, c, and d) can be seen over the JJJ region for circulation Types 1, 2, 3, and 4. In particular, the high pressure system is stronger and broader (Fig. S4a and b), and the seasonal prevailing northerly and northwesterly winds are faster (Fig. S5a and b) over the northwestern part of the JJJ region for Types 1 and 2. This implies that the cold air is more active and the local aerosol pollutants are more easily transported outward under these circulation types. Conversely, there is an obviously smaller northwest-southeast SLP gradient (Fig. S4e, f, g, and h) and weaker seasonal prevailing northwesterly and westerly winds (Fig. S5e, f, g, and h) over the JJJ region for circulation Types 5, 6, 7, and 8. Above all, there is a significant belt of low pressure in the JJJ region, and the seasonal prevailing wind becomes a southwesterly wind in the southeastern part of this region under circulation Types 7 and 8. Such circulations with low pressure and weak wind not only suppress the southward movement of cold air but also promote atmospheric stability in the JJJ region. Consequently, the local aerosol pollutants are prone to accumulating. Therefore, Types 1–4 are defined as the clean circulation types and Types 5–8 are defined as the pollution ones in this study.

Table 3 shows the occurrence frequency of clean and pollution circulation types in winter corresponding to the climatological means and the two types of El Niño years. Compared to the climatological means, it is completely opposite for the composite changes in occurrence frequency of both pollution and clean circulation types between the two types of El Niño years. The total occurrence frequencies of clean and pollution circulation types are reduced by 0.2% and increased by 0.4%, respectively, in the winters of EP El Niño years. By contrast, the corresponding frequencies are increased by 0.6% and decreased by 0.5%, respectively, in the winters of CP El Niño years. These changes imply that the days conducive to the accumulation of local aerosol pollutants are increased in the winters of EP El Niño years, but the opposite occurs in the winters of CP El Niño years. Note that there are some differences among changes in the occurrence frequency of different pollution or clean circulation types. This leads to small magnitudes of the composite changes. However, the WHD anomalies corresponding to EP and CP El Niño years can generally be explained by the composite changes in occurrence frequency of both the pollution and clean circulation types.

In the winters of EP El Niño years, there are negative SLP anomalies over the northwestern and northern parts of the JJJ region but obviously positive SLP anomalies over the southeastern and eastern parts of this region under most circulation types, except for Types 1 and 6 (Fig. 5). Hence, the gradients of SLP are apparently decreased over the JJJ region for each circulation type in the winters of EP El Niño years relative to the climatological means (Fig. S4). Affected by this, southerly wind anomalies occur at the near-surface layer over the JJJ region for both clean and pollution circulation types (Fig. 7c). In addition, the anomalies of SAT over the JJJ region under most circulation types, except for Types 1 and 5, are mainly distributed in the positive anomaly range, indicating that the SAT is generally increased in this region (Fig. 7a). The above analyses show decreased SLP, reduced wind velocity, and increased SAT over the JJJ region under all circulation types in the

winters of EP El Niño years, which lead to a stable synoptic situation. This means that the suppression effects of pollution circulation types on the outward transport of local aerosol pollutants are enhanced over the JJJ region. At the same time, these anomalous synoptic conditions are not conducive to the southward activity of cold air, weakening the diffusion effect of clean circulation types on the local aerosol pollutants in this region.

In contrast, there are positive SLP anomalies over the northwestern and northern parts of the JJJ region, but negative SLP anomalies over the southeastern or southern parts of this region, under the clean circulation types in the winters of CP El Niño years, which increases the northwest-southeast SLP gradient (Fig. S4a–d and 6a–d). Correspondingly, the near-surface meridional wind anomalies over the JJJ region under the clean circulation types are mainly located in the negative anomaly range (Fig. 7d), which means that the seasonal prevailing wind is enhanced in this region. Moreover, the SAT anomalies are also distributed in the negative anomaly range under the clean circulation types (Fig. 7b), indicating a significant decrease in near-surface temperature over the JJJ region. These analyses show that the intensity of synoptic situations conducive to the outward transport of local aerosol pollutants is further enhanced over the JJJ region under the clean circulation types. This may be the reason for the reduction in WHD in this region.

In summary, there are significant differences between the impacts of the two types of El Niño events on the intraseasonal local synoptic conditions. These differences lead to opposite WHD anomalies over the JJJ region in response to different types of El Niño events. In the winters of EP El Niño years, the increase in WHD over the JJJ region may be related to the increased days of pollution circulation types, the decreased days of clean circulation types, the enhanced suppression effect of pollution circulation types on aerosol pollutants, and the weakened diffusion effect of clean circulation types. In the winters of CP El Niño years, the reductions in WHD in the JJJ region are mainly attributable to the increased days and intensity of clean circulation types and the decreased days of pollution circulation types.

## 4 Discussion and conclusions

Based on the long-term site observations of WHD from the CMA, the reanalysis datasets from the ECMWF, and the precipitation reanalysis data from the GPCC, this study explored the impacts of two types of El Niño events on WHD over China's JJJ region and the potential physical mechanisms underlying their differences. The conclusions and discussions are as follows.

The effects of the two types of El Niño events on WHD over the JJJ region are significantly different. There are statistically significant positive (negative) correlation coefficients between WHD over the JJJ region and the Iep (Icp) indices. However, the low correlations likely imply a mild impact of the ENSO on WHD. Correspondingly, WHD increases (decreases) over the JJJ region in the winters of EP (CP) El Niño years. Our results are obviously different from those in previous studies without distinguishing two types of El Niño events (e.g., Li et al., 2017; Sun et al., 2018; Zhao et al., 2018; He et al., 2019), which reported statistically insignificant effects of El Niño on winter haze pollution in North China.

Figure 8 shows the physical mechanisms corresponding to the effects of the EP and CP El Niño on WHD in the JJJ region. The changes in large-scale circulation at the near-surface and mid-low troposphere in East Asia are significantly different in response to the two types of El Niño events, which consequently leads to the opposite effects on WHD over the JJJ region. There are increases in SAT and decreases in SLP over North China in the winters of EP El Niño years. Simultaneously, the seasonal prevailing wind is weakened due to a large range of southerly wind anomalies over the mid-low troposphere in this region. These anomalies suggest that the activity of the EAWM is significantly suppressed and the intensity of cold air is weakened, both of which are conducive to the concurrent increases in WHD over the JJJ region. By contrast, meteorological anomalies, such as near-surface warming and low pressure, are apparent over southern China in the winters of CP El Niño years. The westward shift of the East Asian trough at 500 hPa leads to northerly and northwesterly wind anomalies over the mid-low troposphere in eastern China, which significantly enhances the seasonal prevailing wind. This may result in the

decrease in WHD over the JJJ region during the same period. Furthermore, the positive precipitation anomalies over eastern China are stronger in intensity and wider ranging in the winters of CP El Niño years, which also contributes to the reduction in WHD over the JJJ region.

Our results further indicate an increase in the total occurrence frequency of pollution circulation types and a decrease in that of clean circulation types in the winters of EP El Niño years. These changes support the accumulation and maintenance of local aerosol pollutants in the JJJ region. In addition, there are obvious synoptic condition anomalies, including the reduced SLP gradient, near-surface warming, and weakened northerly wind, over the JJJ region under all circulation types. These changes indicate the enhanced pollution and weakened clean circulation types in the winters of EP El Niño years, which may

be one reason for the increased WHD over the JJJ region. Conversely, the reductions in WHD over the JJJ region are mainly attributable to the increase (decrease) in total occurrence frequency of clean (pollution) circulation types in the winters of CP El Niño years. These anomalous changes result in increased cold air days and thereby facilitate the outward transport of local aerosol pollutants. Meanwhile, the intensity of cold air is enhanced due to the larger SLP gradient, negative temperature anomalies, and stronger near-surface northerly winds over the JJJ region under the clean circulation types. These anomalies

likely contribute to the reduction in WHD in this region.

In recent years, the air quality improvement projects implemented by China's government have effectively controlled the emissions of $PM_{2.5}$ in most areas of China (Zheng et el., 2018; Ding et al., 2019; Gui et al., 2019; Zhang et al., 2019b). However, haze pollution events continue to occur (Zhang et al., 2019a). The impacts of worsening meteorological conditions caused by annual climate change on the haze pollution process are worthy of concern. This study elucidates the potential

physical mechanisms of WHD changes over the JJJ region in response to two types of El Niño events from the perspectives of large-scale circulation and local synoptic condition anomalies. As reported by Yu et al. (2019), we further emphasized the importance of distinguishing the effects of the two types of El Niño events on winter haze pollution in North China. This study has certain implications for further understanding the impact of climate changes on air pollution in China's typical regions. Note that El Niño has the potential to change the composition and size distribution of aerosols by affecting aerosol

transport, deposition, and chemical reactions, which are central to haze pollution (Li et al., 2011; Shaheen et al., 2013; Rajeev et al., 2016; Jayarathne et al., 2018). According to our results, we prefer to attribute the variations in WHD in the JJJ region in El Niño years to the impact of El Niño on the regional transport of aerosols. However, overall aerosol processes related to El Niño could not be precisely characterized at present, as there are few long-term, large-scale observations of aerosol composition, particle types, and size distribution in China. More detailed analyses need to be completed by gathering

more observations and performing more sensitive simulations in future work. In addition, winter haze pollution in China may also be affected by multiple-timescale climate change factors, including the EAWM (Kim et al., 2016), Arctic Oscillation (Chen et al., 2013), Arctic sea ice (Wang and Chen, 2016b), Tibetan Plateau heat source (Xu et al., 2016), and interdecadal variation in snow cover (Yin et al., 2018). Future research should consider how to quantify the comprehensive contributions of different climate change factors to haze pollution in China.


**Data availability**

The monthly haze days dataset can be acquired from http://data.cma.cn/data/cdcindex/cid/6d1b5efbdcbf9a58.html. The monthly Niño3, Niño4, and Niño3.4 indices are available at http://cmdp.ncc-cma.net/download/Monitoring/Index/M_Oce_Er.txt. Daily and monthly ERA-40 and ERA-Interim reanalysis data are

available at https://www.ecmwf.int/en/forecasts/datasets/browse-reanalysis-datasets. The global land surface precipitation data can be acquired from https://climatedataguide.ucar.edu/climate-data.

**Author contributions**

ZW conceived the study. XY, ZW, and HZ performed the analysis and led the manuscript writing. All authors provided

comments and contributed to the text.

**Competing interests**

The authors declare that they have no conflict of interest.

**Acknowledgments**

This study was supported by the key National Natural Science Foundation of China (91644211) and National Key Research and Development Program of China (2016 YFC0203306).

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

**Table 1: The classification of El Niño events**

| Eastern Pacific (EP) | Central Pacific (CP) |
|---|---|
| 1963/1964、1965/1966、1972/1973、1976/1977、1979/1980、1982/1983、1986/1988、1991/1992、1997/1998、2006/2007 | 1968/1970、1977/1978、1994/1995、2002/2003、2004/2005、2009/2010 |







**Table 2: Correlation coefficients between the time series of site-averaged winter haze days in China's JJJ region and different types of El Niño indices. The values in parentheses indicate the correlation coefficients and confidence levels when only considering the stations where the correlations pass a 90% significance level.**

|  | Nino3.4 | Iep | Icp |
|---|---|---|---|
| Cor | 0.04(0.06) | 0.16(0.31) | -0.20(-0.43) |
| P | 0.65(0.45) | 0.05(<0.01) | 0.01(<0.01) |

**Table 3: The occurrence frequencies of each circulation type in winter for climatology and two types of El Niño years (unit: %). The values in parentheses represent changes relative to the climatological means.**

|  |  | climatology | EP El Niño year | CP El Niño year |
|---|---|---|---|---|
| Clean | T1 | 10.3% | 10.6% (+0.3%) | 10.3% (+0%) |
| circulation | T2 | 13.1% | 13.6% (+0.5%) | 14.0% (+0.9%) |
| types | T3 | 14.8% | 14.7% (-0.1%) | 14.9% (+0.1%) |
|  | T4 | 15.6% | 14.7% (-0.9%) | 15.2% (-0.4%) |
|  | Total | 53.8% | 53.6% (-0.2%) | 54.4 % (+0.6%) |
| Pollution | T5 | 17.2% | 16.6% (-0.6%) | 14.9% (-2.3%) |
| circulation | T6 | 13.5% | 13.0% (-0.5%) | 15.1% (+1.6%) |
| types | T7 | 10.6% | 10.2% (-0.4%) | 11.3% (+0.7%) |
|  | T8 | 4.8% | 6.7% (+1.9%) | 4.3% (-0.5%) |
|  | Total | 46.1% | 46.5% (+0.4%) | 45.6 % (-0.5%) |





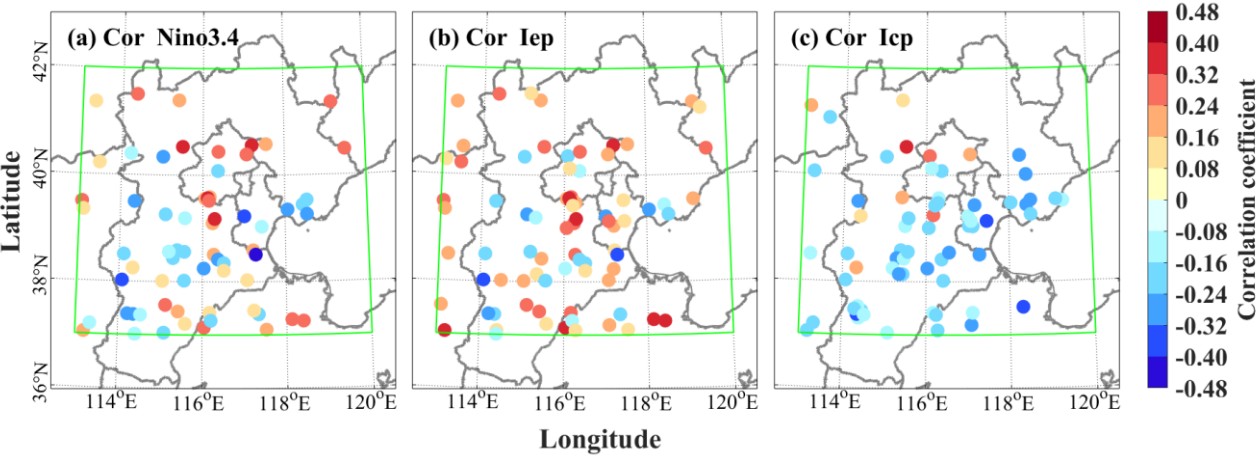

Figure 1: Correlation coefficients between the time series of site-observed winter haze days in JJJ region and (a) I$_{Nino3.4}$, (b) Iep, and (c) Icp indices. The correlations at these sites are significant at 90% confidence level. The green box represents the domain of the JJJ region (37–42°N, 113–120°E) in this study.

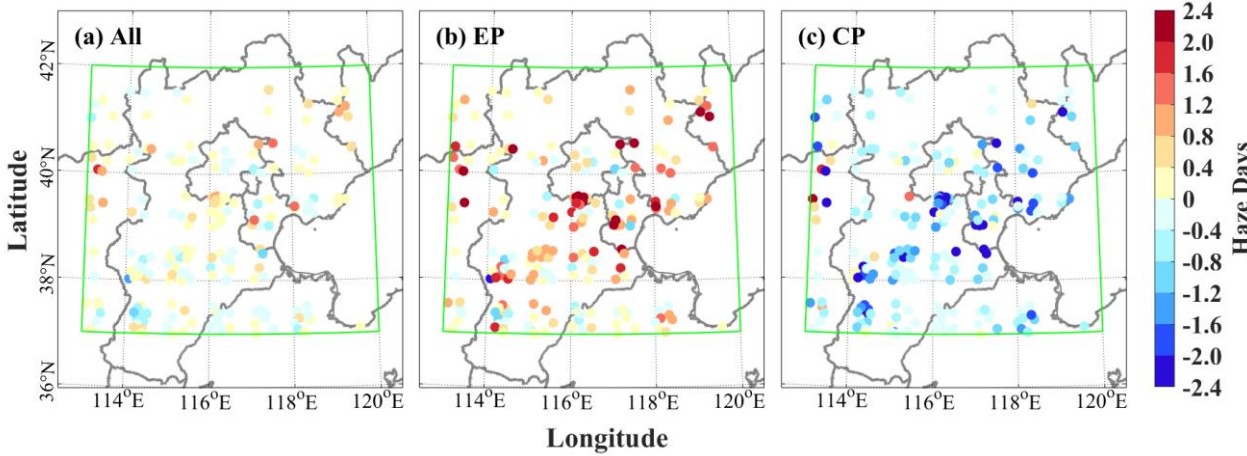

**Figure 2:** Composite changes of winter haze days at all sites over JJJ region in (a) all El Niño, (b) EP El Niño, and (c) CP El Niño years relative to the 1961-2013 mean winter haze days (unit: day). The green box represents the domain of the JJJ region (37–42°N, 113–120°E) in this study.

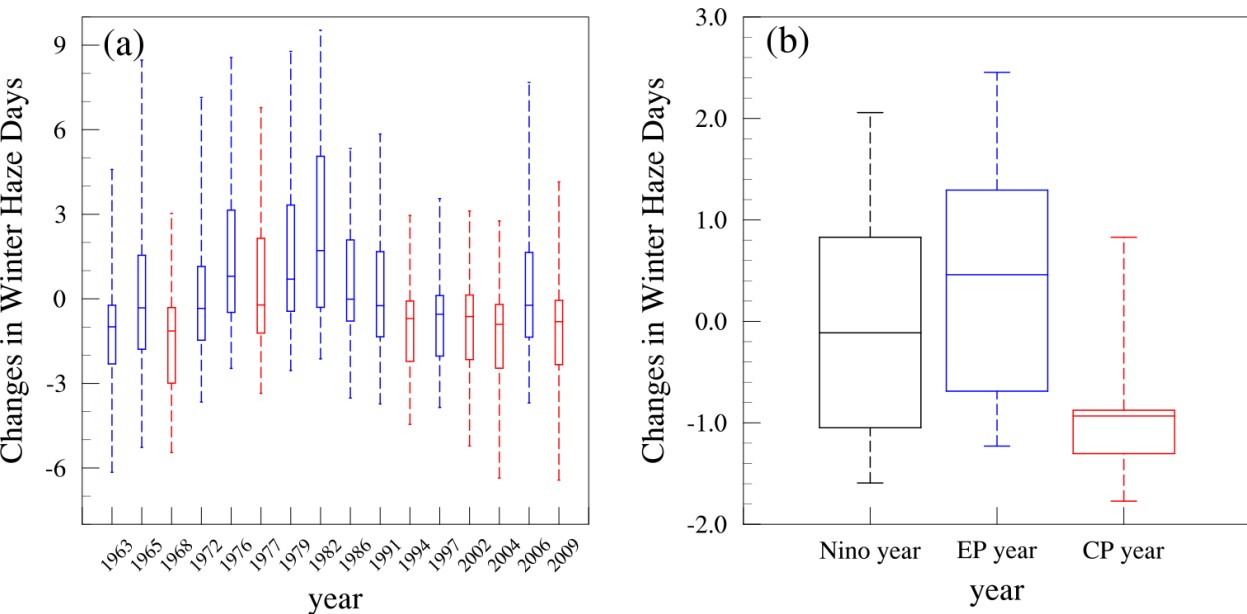

**Figure 3: Box-and-whisker plots of (a) WHD anomalies at all sites over JJJ region in each El Niño year and (b) site-averaged WHD anomalies in different types of El Niño years (unit: day). Each site-averaged WHD anomaly was sampled from a single El Niño year and all these anomalies were divided into groups named as Nino year, EP year, and CP year. The blue, red, and black lines represent the EP, CP, and all El Niño years, respectively. Each box-and-whisker consists of the 5th percentile (the lower point of whisker), 25th quantile (the lower border of box), median (horizontal line in the middle of box), 75th quantile (the upper border of box) and 95th percentile (the upper point of whisker).**

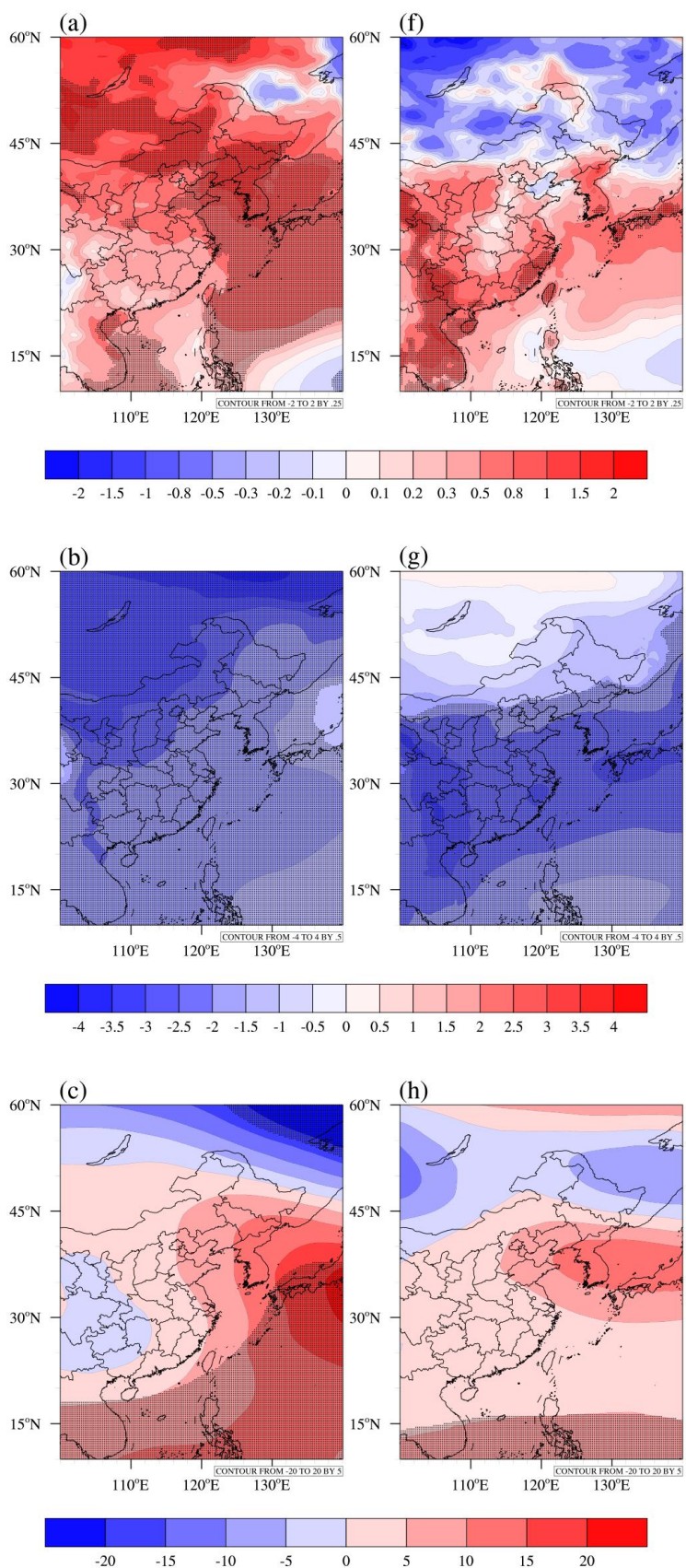

**Figure 4: Winter mean changes in (a, f) air temperature at 2 m (unit: K), (b, g) sea level pressure (unit: hPa), (c, h) geopotential height at 500 hPa (unit: gpm), (d, i) wind averaged from 1000 hPa to 850 hPa (The arrows represent wind vectors and the contours represent wind velocities, unit: m s⁻¹), and (e, j) precipitation (unit: mm) in responses to the two types of El Niño. The left (a-e) and right (f-j) panels represent the differences averaged in 10 EP El Niño and 6 CP El Niño years, respectively, relative to the 1961-2013 climatological means. The dots indicate significance at ≥ 90% confidence level from the t test.**

735

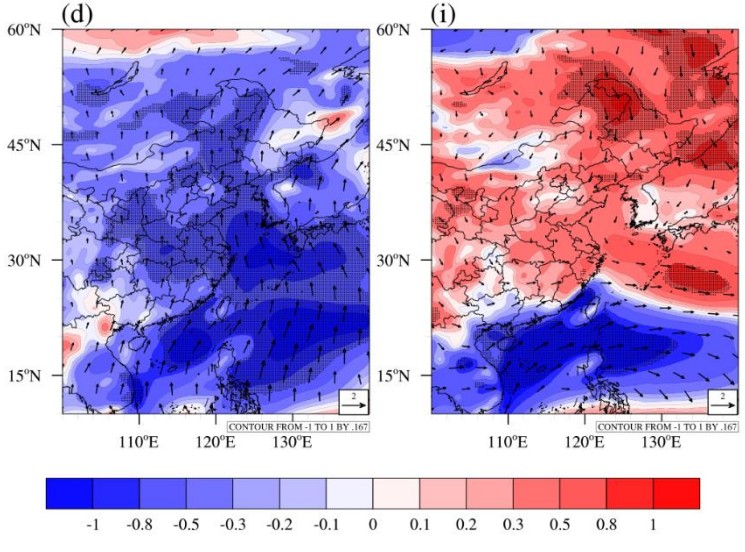

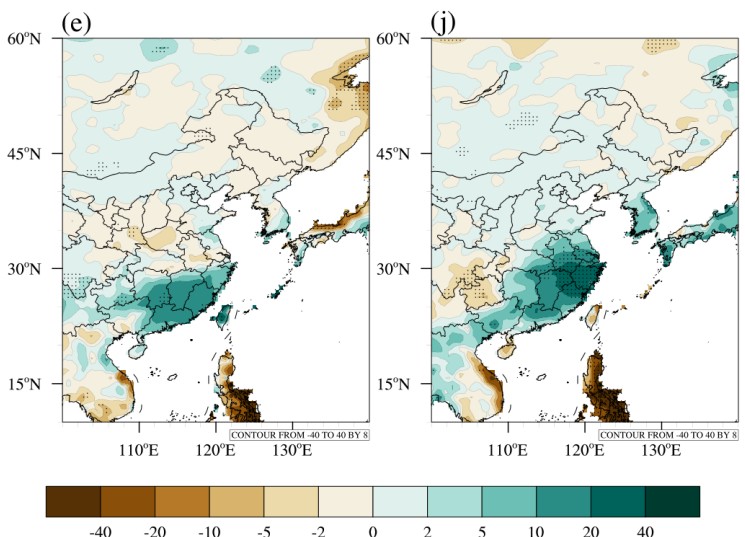

**Figure 4:** *(Continued).*

740

745

750

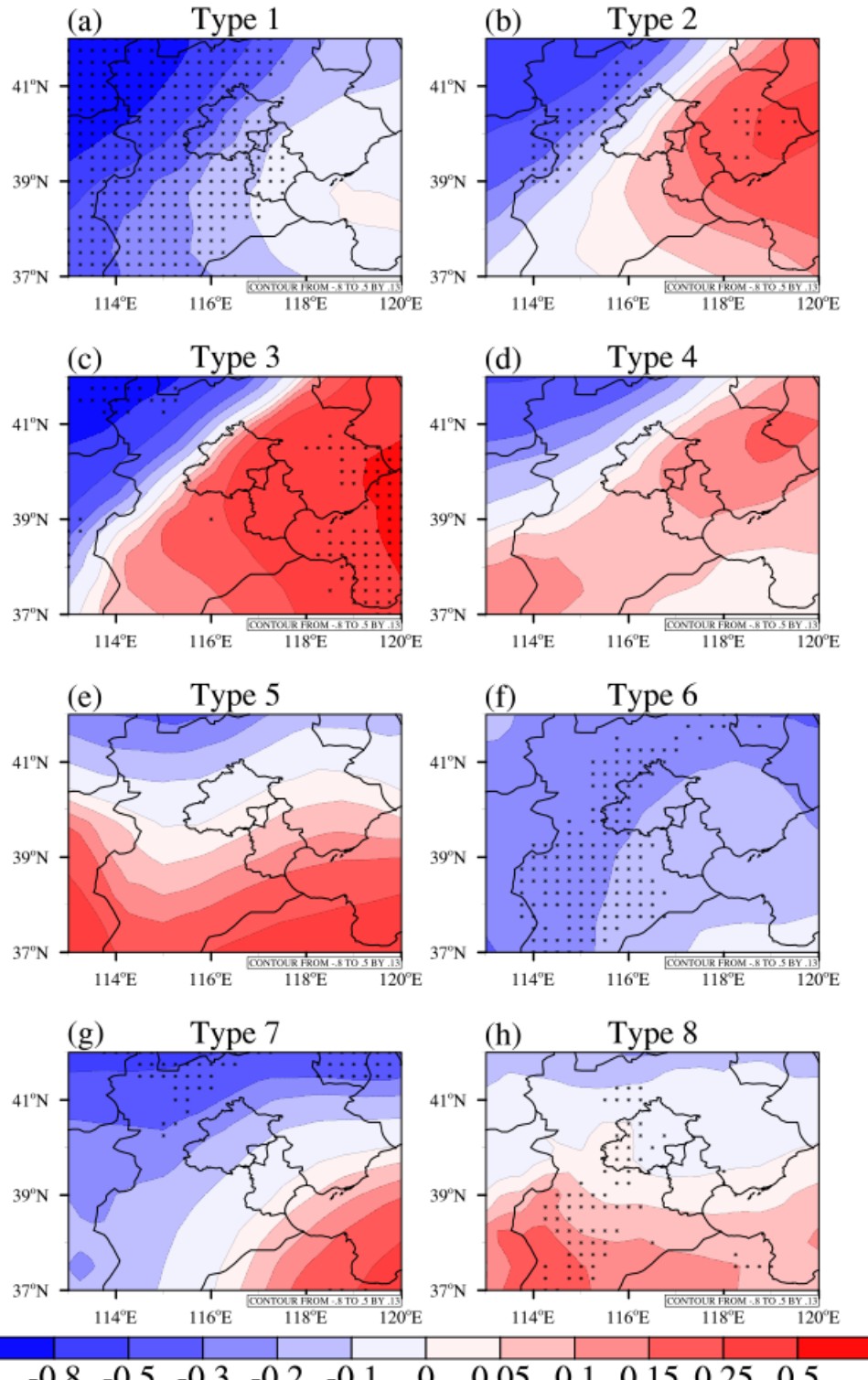

Figure 5: Changes in SLP over JJJ region under eight circulation types in the EP El Niño years relative to the climatological means (unit: hPa). The dots indicate that the differences between more than 60 % of ensemble member pairs have the same sign as the mean differences.

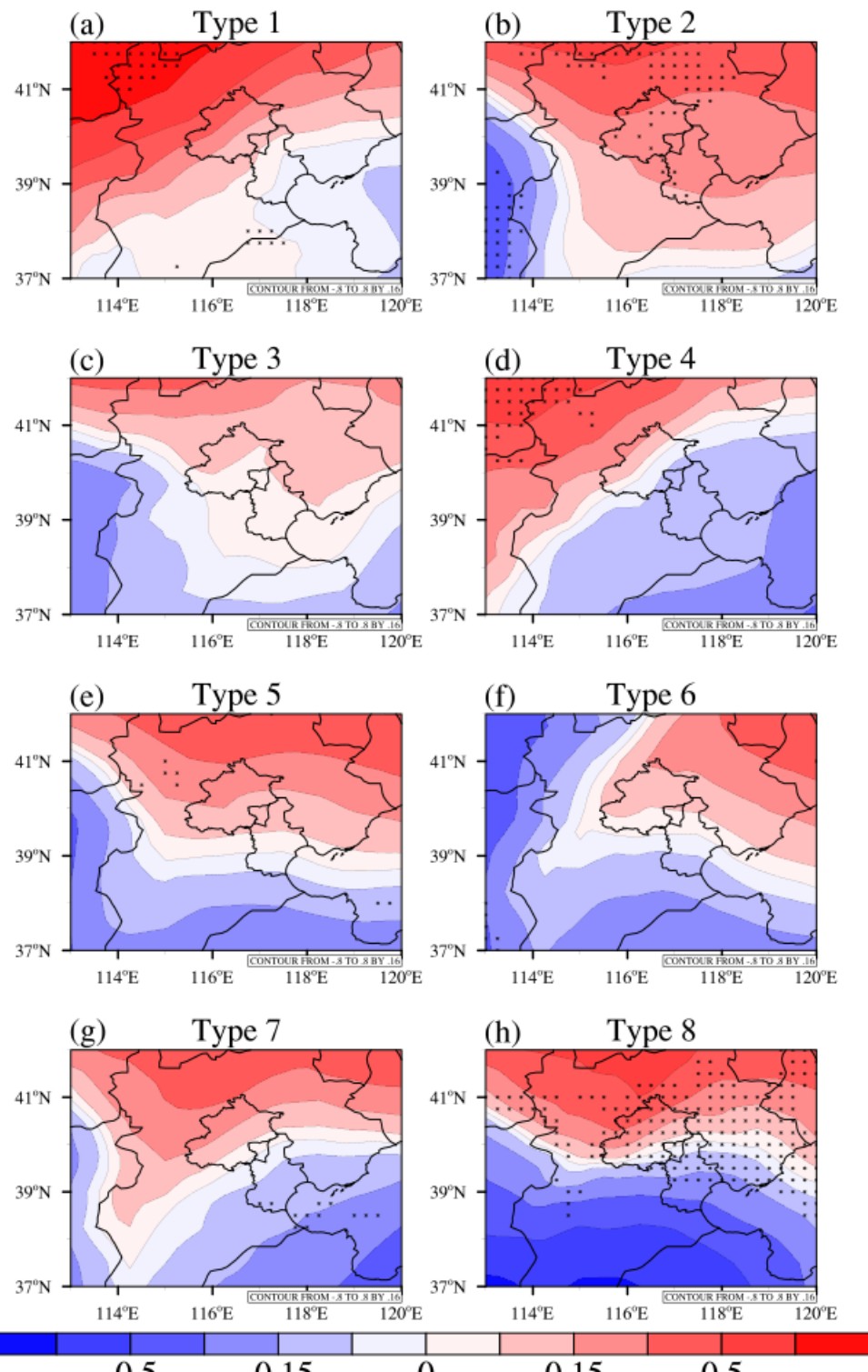

**Figure 6: Changes in SLP over JJJ region under eight circulation types in the CP El Niño years relative to the climatological means (unit: hPa). The dots indicate that the differences between more than 60 % of ensemble member pairs have the same sign as the mean differences.**

760

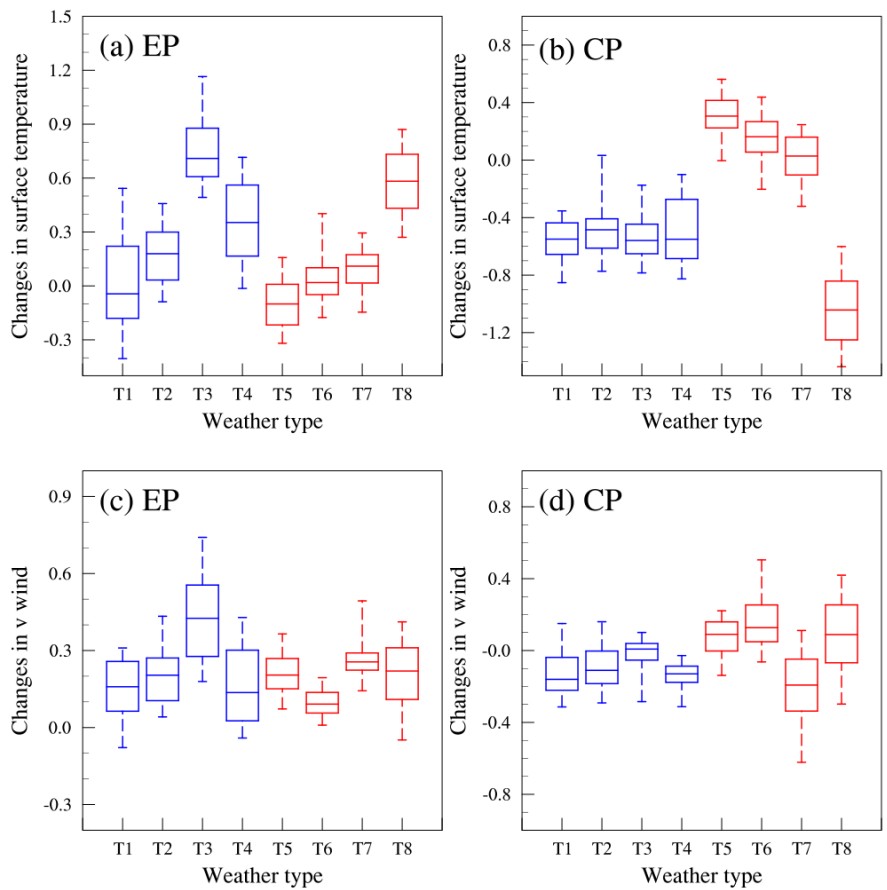

**Figure 7: Box-and-whisker plots of anomalies of (a, b) temperature at 2 m (unit: K) and (c, d) meridional wind at 10 m (unit: m s$^{-1}$) over JJJ region under eight circulation types for different types of El Niño years. The blue and red lines represent the clean and pollution circulation types, respectively. Each box-and-whisker consists of the 5th percentile (the lower point of whisker), 25th quantile (the lower border of box), median (horizontal line in the middle of box), 75th quantile (the upper border of box) and 95th percentile (the upper point of whisker).**

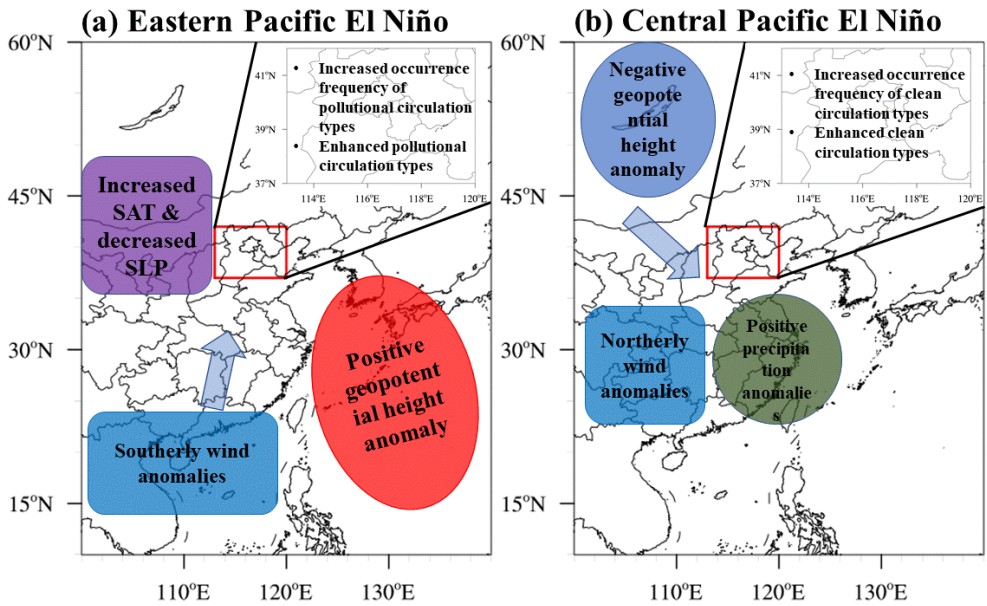

**Figure 8: Schematic diagrams showing the physical mechanisms of effects of (a) Eastern Pacific and (b) Central Pacific El Niño on WHD in JJJ region.**