# Peer review of "Contrasting impacts of two types of El Niño events on winter haze days in China's Jing-Jin-Ji region"

_Atmospheric Chemistry and Physics, 2019_

## Referee Comment (RC1) · Anonymous Referee #1 · 13 Apr 2020

The study by Yu et al. investigates the impact of central (CP) and eastern (EP) Pacific El Niño events on the occurrence of winter haze days (WHD) in the JJJ region in northern China. Based on a statistical analysis of observational data and reanalysis products they conclude that EP El Niños increase the number of WHDs while CP El Niño events decrease their number. Variations in atmospheric circulation patterns over northern China during CP and EP events are suggested to cause this effect.

The manuscript is well structured and presents a thorough analysis on an interesting topic. The presented numbers, however, do not support the claim of a strong effect of the different types of El Niño events on WHDs in the JJJ region (see detailed comments below). I believe that the study is interesting enough to be published but that the conclusions must be formulated much more carefully and worded more in terms of

tendencies. I summarise my concerns and list some specific comments below.

Major concerns:

1) All correlations between the ENSO indices and WHDs as well as changes (e.g. in circulation types) related to ENSO events are very low.

a) Fig. 1 displays pretty low correlations and I have a hard time to believe that they are actually statistically significant when averaged over the region (Tabe 2). How were the degrees of freedom determined for the student t-test? Even if statistically significant, such low correlation values don't argue for a strong impact of ENSO.

b) The response to CP El Niño events appears to be more consistent (judging from the composite change shown in Fig. 2) but the response to EP El Niño events looks rather variable from station to station (Fig. 1 and 2).

c) The box-and-whisker plot (Fig. 3a) indicates that there is quite some spread in the response between individual EP and CP events.

d) From the numbers on the change in circulation types (Table 3 and corresponding text), I would actually conclude that there is hardly any effect of El Niño but maybe I am missing something here?

2) It is stated that the number of haze days changes roughly by 2 during El Niño events (Fig. 2 and corresponding text). How does that compare to the average number of haze days?

3) Regarding the eight circulation types identified in section 3.3 I find it very hard to see the difference between some of them. What determines the number of these types? Since they are grouped together in the following anyway, is it necessary to distinguish between all of them?

Specific comments:

line 9: The first sentence of the abstract sounds strange to me. Maybe use "The El

[Figure]

Niño - Southern Oscillation" instead of just "El Niño".

line 53: What is meant by "Integral El Niño events"?

line 55 to 57: EP and CP El Niños are different flavours of the same climate mode

line 71 to 73: Does this classification differ from other commonly used ENSO classifications?

line 150 to 152: Obviously the averaged correlation values are higher if only values above a certain threshold are considered. I am not sure what to learn from that.

line 190: "worsening meteorological conditions": worse in what respect and compared to what?

line 312: "greatly worth concern" Please rephrase.

---

## Referee Comment (RC2) · Anonymous Referee #2 · 15 Apr 2020

Review comments for "Contrasting impacts of two types of El Niño events on winter haze days in China's Jing-Jin-Ji region" by Yu et al., (2020).

In this study, the authors tele-linked the El Nino events and wintertime haze pollution in Northern China. This study concludes that the occurrence of pollution is connected with El Nino modes. Generally speaking, the paper can be significantly improved with the inclusion of chemical research and discussions when dealing with the haze topic (e.g., the composition and the response by each species). This study is more like a purely statistical analysis with insufficient mechanism explanations. Moreover, the overall structure of this paper is somewhat mixed up and the English of this study needs some improvements. I have the following concerns before the formal publication of this study.

[Figure]

Specific comments:

1. This study emphasized haze days, however, without specifying the source of haze. For example, the chemistry here should definitely be discussed. Is PM the one to blame? If so, what is the composition? Also, when conducting the correlation analyses, what are the correlation to individual particle types? Any size distribution biases?

2. In this study, the Nino data used were provided by CMA. I am wondering what is the difference between the CMA Nino data and NOAA nino data? Authors should give more in-depth descriptions on the products they use.

3. The authors heavily relied on the ERA data for both ERA-40 and ERA-interim. Why not using the latest EAR5 data instead? I understand the ERA-40 is for older records but the ERA5 should be available for more recent years. Using state-of-art products boost the innovative part of this study.

4. The authors should expand section 2.3. The described method was very generic and details-lacking. It is very hard for readers to comprehend what has been done. Also, the first two paragraphs of 3.1 should be placed in the method section instead of the results.

5. It is hard to tell whether the correlation results shown in Figure 1 are significant or not as the highest correlation is around 0.5 for both positive and negative correlations. Can authors please justify the significance of these correlations numbers?

6. The caption of Figure 5 "The dots indicate that the differences between more than 60

7. Since this paper primarily focuses on the JJJ region, I would recommend authors to highlight the boundary of this region when making the plots, especially in zoomed-in cases (e.g., Figures 1, 2 and 5).

8. Authors, please check the right panel of Figure 3 for $CP_year. The lower whisky overlaps with 25$

9. This paper discusses the positive precipitation anomaly for the CP case. How about the precipitation for the EP case?

10. In Figure 1, I noticed one dot has distinctive signs between nino 3.4 and EP, shown below in square. Why is that the case? I assume these two regions shall be pretty close.
* * *
[Figure]

[Figure]

**Fig. 1.**

---

## Author Comment (AC1) · 9 Jun 2020

**Author's Response to Anonymous Referee 1**

**Anonymous Referee #1:**

The study by Yu et al. investigates the impact of central (CP) and eastern (EP) Pacific El Niño events on the occurrence of winter haze days (WHD) in the JJJ region in northern China. Based on a statistical analysis of observational data and reanalysis products they conclude that EP El Niño increase the number of WHDs while CP El Niño events decrease their number. Variations in atmosphere circulation patterns over northern China during CP and EP events are suggested to cause this effect. The manuscript is well structured and presents a thorough analysis on an interesting topic. The presented numbers, however, do not support the claim of a strong effect of the different types of El Niño events on WHDs in the JJJ region (see detailed comments below). I believe that the study is interesting enough to be published but that the conclusions must be formulated much more carefully and worded more in terms of tendencies. I summarise my concerns and list some specific comments below.

Reply: Thanks for the referee's positive comments and constructive suggestions. We have taken the referee's comments into consideration and carefully revised the manuscript. Please see our detailed point-by-point reply below.

Major concerns:

Q1: All correlations between the ENSO indices and WHDs as well as changes (e. g. in circulation types) related to ENSO events are very low.

a)  Fig.1 displays pretty low correlations and I have a hard time to believe that they are actually statistically significant when averaged over the region (Table 2). How were the degrees of freedom determined for the student t-test? Even if statistically significant, such low correlation values don't argue for a strong impact of ENSO.
    Reply: We sampled the monthly data in each winter (December, January, and February) from 1961 to 2012, which means that the degree of freedom is 154. According to the threshold table of correlation coefficient (Zhou and Zheng, 1997), the absolute values of correlation coefficients between the site-averaged WHDs and EP and CP El Niño indices (Table 2) are larger than the threshold of 0.154. This indicates that they are statistically significant. Compared to the single site-observed WHDs, the correlations between the site-averaged WHDs and El Niño indices are easier to pass the significance test because the disturbances of local emissions, urbanization, and topography on WHDs may be eliminated. In addition, the correlation analysis is the primary step. We further elaborated the opposite impacts of two types El Niño events on the WHD in the JJJ region via exhibiting the differences between the changes in WHDs in two type El Niño years (Figures 2 and 3).
    We agree with you that the low correlation values may not argue for a strong impact of ENSO. We have supplemented such description. Please see Lines 156-157 and 311-313 in the revised manuscript.
    References:
    Zhou, Y. H. and Zheng, D. W.: A new calculation method of correlation coefficient test table. Annals of Shanghai Observatory Academia Sinica, 18, 18-23, 1997.

b)  The response to CP El Niño events to be more consistent (judging from the composite change shown in Fig. 2) but the response to EP El Niño events looks rather variable from station to station (Fig. 1 and 2).

Reply: Haze pollution is a sophisticated problem because it includes the comprehensive impacts of emissions, weather conditions, and even topography. This may lead to some biases of correlation coefficients among different sites when examining the correlations between the single-site WHDs and El Niño indices. But we can still find the consistently composite positive WHD anomalies, corresponding to EP El Niño years, at most sites (149 sites; accounting for 76.4% of all sites) in the JJJ region in Fig. 2b. In addition, the positive correlation between EP El Niño events and WHDs is also illustrated at 121 stations (accounting for 62.1% of all stations; Fig. 1b). The corresponding proportion will increase to 70.5% if we only consider the stations at which the correlations pass a significance level of 90%. We have supplemented quantitative descriptions to highlight the positive correlations between the WHDs and EP El Niño events. Please see Lines 147-154 and 161-164 in the revised manuscript.

c) The box-and-whisker plot (Fig. 3a) indicates that there is quite some spread in the response between individual EP and CP events.

Reply: The aerosol pollutants in the JJJ region are not only subject to the interannual change of emissions and the multiple-time scale climate changes as we discussed in the section 4, but also affected by the variations in local emissions, urbanization, and topography. These result in differences in distributions of WHD anomalies among individual EP and CP El Niño years and some spread of WHD anomalies in special El Niño year. However, we do find out the variations of WHD anomalies in phase with individual EP and CP El Niño years (Fig. 3a), which are characterized by a larger distribution in the positive range in response to most of EP El Niño years, but a larger distribution in the negative range in response to most of CP El Niño years. The site-averaged results also indicate the opposite distribution of changes in WHD corresponding to EP and CP El Niño years (Fig. 3b). We have supplemented more explanation about the box-and-whisker plot to elaborate our results. Please see Lines 171-173 in the revised manuscript.

d) From the numbers on the change in circulation types (Table 3 and corresponding text), I would actually conclude that there is hardly any effect of El Niño but maybe I am missing something here?

Reply: Indeed. These composite changes in the occurrence frequency of both pollution and clean circulation types seem to be small corresponding to EP and CP El Niño years. However, it is more complicated when we examine those changes for each circulation type. For example, in the winter of EP El Niño years, the changes in the occurrence frequency of the pollution (clean) circulation types range from -0.53% (-0.95%) to 1.97% (0.45%). The corresponding changes range from -2.33% (-0.4%) to 1.55% (0.84%) in the winter of CP El Niño years. The values may be small when calculating the composite changes. In addition, it is more difficult to further link the changes in WHDs to variation of a specific circulation type due to the lack of long-term daily haze pollution data. But we can generally explain the WHD anomalies in EP and CP El Niño years according to the composite changes in occurrence frequency of both pollution and clean circulation types. Finally, there is no formula currently to quantitatively describe the relationship between synoptic-scale circulation anomalies and haze pollution. More detailed works are needed in the future. We have supplemented more details in Table 3 and corresponding text. Please see Lines 268-277 in the revised manuscript.

Q2: It is stated that the number of haze days changes roughly by 2 during El Niño events (Fig. 2 and corresponding text). How does that compare to the average number of haze days?

Reply: The changes of 2 haze days account for 17% to 79% and -13% to -70% of the average numbers of haze days, respectively, at these sites. We have added the figure of percentages of WHD anomalies in all El Niño, EP El Niño, and CP El Niño years, respectively, relative to the climatological means (new Figure S2). We have also supplemented some descriptions in the revised manuscript. Please see Lines 161-164 in the revised manuscript.

Q3: Regarding the eight circulation types identified in section 3.3 I find it very hard to see the difference between some of them. What determines the number of these types? Since they are grouped together in the following anyway, is it necessary to distinguish between all of them?

Reply: We have supplemented the difference in each circulation type in Table 3. In this study, we used the K-means clustering algorithm to classify different circulation types. The optimal number of classifications is determined by the inflection point of criterion function. More detail descriptions are reported in Liu and Gao (2011). There remain some differences in the patterns of circulation anomalies among different pollution or clean circulation types in Figures 5, 6 and 7, although their common variations can be used to explain the WHD anomalies in response to EP and CP El Niño years. The differences may further relate to the intraseasonal changes in haze pollution. However, as answered in Q1 (d), exploring this link needs long-term daily haze pollution data, which is lacking at present.

References:

Liu, D. and Gao, S. C.: Determining the number of clusters in K-means clustering algorithm. Silicon Valley, 6: 38-39, 2011.

Specific comments:

Q4: line 9: The first sentence of the abstract sounds strange to me. Maybe use "The El Niño-Southern Oscillation" instead of just "El Niño".

Reply: We have changed this sentence into "El Niño is complicated due to its diverse distribution features and intensities.". Please see Line 9 in the revised manuscript.

In this study, we only focus on the impacts of El Niño, but not the La Niña. Therefore, it may be more accurate to only mention "El Niño" here.

Q5: line 53: What is meant by "Integral El Niño events"?

Reply: The "Integral El Niño events" used here means the overall El Niño events without classification. We have revised it. Please see Line 53 in the revised manuscript.

Q6: line 55 to 57: EP and CP El Niño are different flavours of the same climate mode

Reply: It may be an ambiguity here. We agree with you that both types of El Niño events are the warm conditions of El Niño–Southern Oscillation—the leading climate mode of interannual variability in the tropical Pacific. However, we want to emphasize two dynamic modes of quasi-quadrennial and quasi-biennial oscillations at here. These independent modes coexist in the tropical Pacific and may modulate the features of ENSO due to their interannual and interdecadal changes (Bejarano et al., 2008; Wang and Ren, 2017). Their interplay also contributes to the spatial diversity of the observed ENSO events, like EP and CP El Niño (Timmermann et al., 2018). We have revised the manuscript to avoid the ambiguous expression. Please see Lines 55-59 in the revised manuscript.

References:

Bejarano, L. and Jin, F. F.: Coexistence of equatorial coupled modes of ENSO. Journal of Climate, 21(12): 3051-3067, doi:10.1175/2007jcli1679.1, 2008.

Timmermann, A., An, S. I., Kug, J. S., Jin, F. F., Cai, W. J., Capotondi, A., Cobb, K., Lengaigne, M., McPhaden, M. J., Stuecker, M. F., Stein, K., Wittenberg, A. T., Yun, K. S., Bayr, T., Chen, H. C., Chikamoto, Y., Dewitte, B., Dommenget, D., Grothe, P., Guilyardi, E., Ham, Y. G., Hayashi, M., Ineson, S., Kang, D., Kim, S., Kim, W. M., Lee, J. L., Li, T., Luo, J. J., McGregor, S., Planton, Y., Power, S., Rashid, H., Ren, H. L., Santoso, A., Takahashi, K., Todd, A., Wang, G. M., Wang, G. J., Xie, R. H.,

Yang, W. H., Yeh, S. W., Yoon, J., Zeller, E., and Zhang, X. B.: El Niño–Southern Oscillation complexity. Nature, 559(7715): 535-545, doi:10.1038/s41586-018-0252-6, 2018.

Wang, R. and Ren, H. L.: The linkage between two ENSO types/modes and the interdecadal changes of ENSO around the year 2000. Atmospheric and Oceanic Science Letters, 10(2): 168-174, doi:10.1080/16742834.2016.1258952, 2017.

Q7: line 71 to 73: Does this classification differ from other commonly used ENSO classifications?

Reply: Yes, it does. This classification is based on an equation set that contain two ENSO indexes (Niño 3 and Niño 4 index) as shown in section 2.2, although it still employs the common monitoring areas of sea surface temperature anomalies. At present, there are quite some differences among the monitoring results for the same ENSO event due to employ differently single index, like the Japan Meteorological Agency (JMA) index, the Multivariate ENSO Index (MEI), and the El Niño Modoki Index (EMI). The classification used in this study shows the better performance in monitoring different types of historical ENSO events and determining the characteristics of the ENSO events than any single index. This has been reported in previous studies (Cao et al., 2013; Ren et al., 2017).

References:

Cao, L., Sun, C. H., Ren, F. M., Yuan, Y. and Jiang, J.: STUDY OF A COMPERHENSIVE MONITORING INDEX FOR TWO TYPES OF ENSO EVENTS. Journal of Tropical Meteorology, 29(1):66-74, 10.3969/j.issn.1004-4965.2013.01.008, 2013.

Ren, H. L., Sun, C. H., Ren, F. M., Yuan, Y., Lu, B., Tian, B., Zuo, J. Q., Liu, Y., Cao, L, Han, R. Q., Jia, X. L. and Liu, C. Z.: Identification method for El Niño/La Niña events. The People's Republic China's National Standard GB/T 33666-2017, May 2017. Beijing: Standards Press of China, 1-6, 2017.

Q8: line 150-152: Obviously the averaged correlation values are higher if only values above a certain threshold are considered. I am not sure what to learn from that.

Reply: As mentioned in Q1, the variations in local emissions, urbanization, and topography make haze pollution in the JJJ region more complicated. These factors not only lead to some biases of correlation coefficients among different sites when examining the correlations between the single-site WHDs and El Niño indices, but also contribute to the lower correlations between the site-averaged WHD and El Niño indices. Therefore, we selected the stations at which the correlations pass a significance level of 90% and calculated the correlations between different El Niño indices and the averaged WHDs at these stations. These higher correlation coefficients of 0.31 and -0.43, corresponding to EP and CP El Niño events, further illustrate the significant opposite impacts of two types El Niño on the WHDs in the JJJ region.

Q9: line 190: "worsening meteorological conditions": worse in what respect and compared to what?

Reply: As reported in the previous studies (Chen et al., 2015; Chang et al., 2016), the lower sea surface pressure and the higher surface air temperature in the northeastern Eurasia are favorable for the maintenance and development of atmospheric pollutant and defined as the worsening meteorological conditions (Zhang et al., 2018). In this study, the above anomalies are apparent in the winter of EP El Niño years.

References:

Chang, L. Y., Xu, J. M., Tie, X. X., and Wu, J. B.: Impact of the 2015 El Nino event on winter air quality in China. Scientific Reports, 6(1):34275, doi:10.1038/srep34275, 2016.

Chen, H. P. and Wang, H. J.: Haze Days in North China and the associated atmospheric circulations based on daily visibility data from 1960 to 2012. Journal of Geophysical Research Atmospheres, 120(12):5895-5909, doi:10.1002/2015jd023225, 2015.

Zhang, X. Y., Zhong, J. T., Wang, J. Z., Wang, Y. Q., and Liu, Y. J.: The interdecadal worsening of weather conditions affecting aerosol pollution in the Beijing area in relation to climate warming. Atmospheric Chemistry and Physics, 18(8), 5991-5999,

doi:10.5194/acp-18-5991-2018, 2018.

Q10 line 312: "greatly worth concern" Please rephrase.

Reply: We have changed the description here into "The impacts of worsening meteorological conditions caused by annual climate change on the haze pollution process are worthy of concern.". Please see the Lines 343-344 in the revised manuscript.

---

## Author Comment (AC2) · 9 Jun 2020

**Author's Response to Anonymous Referee 2**

**Anonymous Referee #2:**

Review comments for "Contrasting impacts of two types of El Niño events to winter haze days in China's Jing-Jin-Ji region" by Yu et al., (2020).

In this study, the authors tele-linked the El Niño events and wintertime haze pollution in Northern China. This study concludes that the occurrence of pollution is connected with El Niño modes. Generally speaking, the paper can be significantly improved with the inclusion of chemical research and discussion when dealing with the haze topic (e.g., the composition and response by each species). This study is more like a purely statistical analysis with insufficient mechanism explanations. Moreover, the overall structure of this paper is somewhat mixed up and the English of this study needs some improvements. I have the following concerns before the formal publication of this study.

Reply: Thank you very much for the thorough comments and suggestions. These comments and suggestions are very helpful to improve the quality of the manuscript. We have made revisions according to these comments. Please find the following point-point reply. In addition, the English of the manuscript has been improved by native speakers of English. For a certificate, please see: http://www.textcheck.com/certificate/fUwXLd

Specific concerns:

Q1: This study emphasized haze days, however, without specifying the source of haze. For example, the chemistry here should definitely be discussed. Is PM the one to blame? If so, what is the composition? Also, when conducting the correlation analyses, what are the correlation to individual particle types? Any size distribution biases?

Reply: Thanks a lot for these helpful comments. We agree with you that anomalous weather conditions may affect the chemistry of some aerosol types, such as sulfate and nitrate. However, the haze days defined by visibility and relative humidity is the only available long-term observation data that reflects air pollution levels in China. There are few long-term large-scale observations of aerosol composition, particle types, and size distribution in China for the correlation analyses. More detailed analyses need to be solved by gathering more observations and performing some sensitive simulations in future work. Alternatively, we added more analyses and discussions to further illustrate that the variations of WHDs in the JJJ region in response to EP and CP El Niño years are more attributed to the regional transport of aerosol pollutants caused by two types of El Niño. Please see the last paragraph of both section 3.2 and section 4 in the revised manuscript.

Q2: In this study, the Niño data used were provided by CMA. I am wondering what is the difference between the CMA Nino data and NOAA nino data? Authors should give more in-depth descriptions on the products they use.

Reply: The definitions of the Niño indices between CMA and NOAA are the same. Referring to the People's Republic China's National Standard (Ren et al., 2017), the Niño indices provided by CMA are calculated using the Hadley Centre Sea Ice and Sea Surface Temperature Data (HadISST) from March 1961 to December 1981 and the National Oceanic and Atmospheric Administration (NOAA) daily optimum interpolation (OI.v2) SST dataset from January 1982 to February 2013. We have supplemented these descriptions. Please see Lines 89-92 in the revised section 2.1.
References:
Ren, H. L., Sun, C. H., Ren, F. M., Yuan, Y., Lu, B., Tian, B., Zuo, J. Q., Liu, Y., Cao, L, Han, R. Q., Jia, X. L. and Liu, C. Z.:

Identification method for El Niño/La Niña events. The People's Republic China's National Standard GB/T 33666-2017, May 2017. Beijing: Standards Press of China, 1-6, 2017.

Q3: The authors heavily relied on the ERA data for both ERA-40 and ERA-interim. Why not using the latest ERA5 data instead? I understand the ERA-40 is for older records but the ERA5 should be available for more recent years. Using state-of-art products boost the innovative part of this study.

Reply: Thanks a lot for this advice! We have reexamined our results using the latest ERA5 data and compared them with our original results. In the new results, the data from March 1961 to December 1978 are derived from the ERA-40 reanalysis data and that from January 1979 to February 2020 are derived from the ERA5 reanalysis data. As seen in Figures A1a and e, there are increases in surface air temperature (SAT) over northern China in the winters of EP El Niño years, but corresponding decreases in the winters of CP El Niño years, which are more obvious than our original results. Similar to Figures A1a and e, the opposite patterns of sea level pressure (SLP) anomalies over northern China in response to two types of El Niño years are shown in Figures A1b and f, although these anomalies are weaker than our original results. The patterns of geopotential height anomalies at 500 hPa corresponding to the EP and CP El Niño years are generally consistent with our original results. But the resulting changes in wind over northern China in the winter of both EP and CP El Niño years are weaker for the new data. For the changes in intraseasonal atmospheric circulation in each circulation type in response to two types of El Niño years, there are some differences between the new and original results (Figures A2, 3, and 4). However, the new results still capture the decreased SLP gradients, the southerly wind and positive SAT anomalies in most of circulation types in the winters of EP El Niño years over the JJJ region (Figures A2, A4a and e). Meanwhile, the opposite anomalies, such as increased SLP gradients, northerly wind, and negative SAT anomalies, corresponding to the CP El Niño years are shown in the new results (Figures A3, A4b and d). In brief, the new results also clearly show the differences of atmospheric circulations corresponding to two types of El Niño years at both interannual and interdecadal timescales. These are in line with our original analyses, so we didn't replace the data in this manuscript.

[Figure]

[Figure]

**Figure A1: Winter mean changes in (a, e) air temperature at 2 meter (unit: K), (b, f) sea level pressure (unit: hPa), (c, g) geopotential height at 500 hPa (unit: gpm), and (d, h) wind averaged from 1000 hPa to 850 hPa (The arrows represent wind vectors and the contours represent wind velocities, unit: m s⁻¹) in responses to the two types of El Niño. The left (a-d) and right (e-h) panels represent the differences averaged in 11 EP El Niño and 7 CP El Niño years, respectively, relative to the 1961-2020 climatological means. The dots indicate significance at ≥ 90% confidence level from the t test.**

[Figure]

**Figure A2: Same as Figure 5, but using the new data set.**

[Figure]

**Figure A3: Same as Figure 6, but using the new data set.**

[Figure]

**Figure A4: Same as Figure 7, but using the new data set.**

Q4: The authors should expand section 2.3. The described method was very generic and details-lacking. It is very hard for readers to comprehend what has been done. Also, the first two paragraphs of 3.1 should be placed in the method section instead of the results.

Reply: Accepted. We have revised the section 2.3 to make sure that readers can clearly understand what has been done. In addition, we have moved the first paragraph of section 3.1 to section 2 as a separate section 2.4. The second paragraph of section 3.1 includes more results of correlation analysis, so we remain it in the original section. Please see the revised section 2.3, 2.4, and 3.1.

Q5: It is hard to tell whether the correlation results shown in Figure 1 are significant or not as the highest correlation is around 0.5 for both positive and negative correlations. Can authors please justify the significance of these correlation numbers?

Reply: All the correlation results shown in Figure 1 are significant at 90% confidence level. We have revised the figure caption. Please see the revised manuscript.

Q6: The caption of Figure 5 "The dots indicate that the differences between more than 60

Reply: We guess that the referee wants to know more details about the dots. We sampled daily data corresponding to each of synoptic-scale circulation types in 10 EP El Niño years, and then calculated the differences by subtracting the 1961-2013 climatological averaged result of each types from them. The dots indicate that more than 60% of all the differences have the same sign as the mean differences. This approach to some extent represents the statistics significance of the results.

Q7: Since this paper primarily focuses on the JJJ region, I would recommend authors to highlight the boundary of this region when making the plots, especially in zoomed-in cases (e.g., Figures 1, 2 and 5).

Reply: We agree entirely with the referee's view. We have highlighted our research domain of the JJJ region with the green boxes in Figures 1, 2 and S2. Note that the areas shown in Figures 5 and 6 are entirely consistent with our research domain.

Q8: Authors, please check the right panel of Figure 3 for $CP_{year}$. The lower whisky overlap with 25.

Reply: Thank you for your reminding. We have replaced the winter mean data with three monthly data sampled from each El Niño winter and replotted this figure. In addition, we have also replaced the extremums with the 5th and 95th percentile. Please see the revised manuscript.

Q9: This paper discusses the positive precipitation anomaly for the CP case. How about the precipitation for the EP case?

Reply: As seen in Figure 4e, the monthly precipitation is generally increased over southern China in the winters of EP El Niño years, with the maximum changes exceeding 10 mm. But there are slightly negative anomalies of precipitation over central and northeastern China. The area with positive precipitation anomalies over the JJJ region is smaller in EP El Niño years compared to that in CP El Niño years, although a comparable increase in precipitation over this region occurs with both types of El Niño years. We have increased the description of precipitation anomalies in response to EP El Niño years. Please see Lines 225-226 in the revised manuscript.

Q10: In Figure 1, I noticed one dot has distinctive signs between nino 3.4 and EP, shown below in square. Why is that case? I assume these two regions shall be pretty close.

Reply: Two stations with close distance may belong to urban and rural areas, respectively. This can lead to a distinct difference in their underlying surface and local emissions. In addition, two stations with close distance may differ greatly in altitude due to the complicated terrain. The above factors will complicate haze pollution and may be the reason for the distinctive difference in signs between two adjacent stations.

---

## Author Response (AR2)

**Author's Response to Anonymous Referee 2**

**Anonymous Referee #2:**

Review for revised version of "Contrasting impacts of two types of El Niño events to winter haze days in China's Jing-Jin-Ji region" by Yu et al.

The authors have adequately addressed most of my concerns with the previous version of their manuscript. I recommend publication with some minor additional changes listed below.

Reply: We gratefully thank the referee for the constructive suggestions for our manuscript. These suggestions guided us further to improve the manuscript. We provide point-by-point responses below.

Q1: Regarding the statistical significance of the correlations shown in Fig. 1 and Table 2, it is stated in the response to the review that the degrees of freedom were determined from the number of months taken into account (DJF from 1961 to 2012).

Are the three consecutive winter months independent of each other? Otherwise the degrees of freedom would reduce to 50 and the threshold for significance would increase to something like 0.25.

However, I think the paper would benefit from not putting too much emphasis on these numbers anyway. As the authors elude to in the revised version, there are a number of local factors complicating the relationship between El Niño events and WHD. I think it would thus be more instructive to focus on the consistent responses found and formulate them in terms of tendencies rather than first stating that some very low correlations are significant and then admitting that they are weak.

To give an example, instead of writing "As seen in Figure 1, the absolute values of the correlation coefficients at some stations exceed 0.4. There are statistically significant correlations between the site averaged WHD in the JJJ region and the Iep and Icp indices (p≤0.05), with correlation coefficients of 0.16 and 0.2, respectively (Table 2). These low correlation values likely imply a mild impact of ENSO on WHD. However, the corresponding correlation coefficients reach 0.31 and 0.43, respectively, with a confidence level of 99%, when only considering the stations at which the correlations pass a significance level of 90%" (lines 154 to 159) one could say something like "Even though the correlations are rather low with values hardly exceeding at 0.4 at individual stations and correlations of WHD averaged over the JJJ region and Iep and Icp of only 0.16 and -0.2, respectively, a pattern emerges with central Pacific El Niños being associated with less WHD and a tendency for more WHD during eastern Pacific El Niños. This is corroborated by the composite analysis…"

Reply: Thank a lot for these instructive comments. We have improved the description of results of correlation analysis in the first paragraph of section 3.1. Please see Lines 146-157 in the revised manuscript.

Specific comments:

Q2: Line 9: I would suggest "El Niño events differ widely in their patterns and intensities"
Reply: Done. Please see Line 9 in the revised manuscript.

Q3: Line 10; "events" instead of "event"
Reply: Done. Please see Line 10 in the revised manuscript.

Q4: Line 17: "obvious" or "clear instead of "obviously"

Reply: Done. Please see Line 17 in the revised manuscript.

Q5: Line 39: "As the strongest" instead of "As a strongest"

Reply: Done. Please see Line 39 in the revised manuscript.

Q6: Line 106: What is meant by "smoothing average"? A running mean?

Reply: Yes, it refers to "running mean" at here. We have replaced "smoothing average" with "running mean". Please see Line 106 in the revised manuscript.

**A list of all relevant changes made in the manuscript in response to Referee 2**

[revised manuscript text omitted]